# Links between Soil Bacteriobiomes and Fungistasis toward Fungi Infecting the Colorado Potato Beetle

**DOI:** 10.3390/microorganisms11040943

**Published:** 2023-04-04

**Authors:** Ekaterina Chertkova, Marsel R. Kabilov, Olga Yaroslavtseva, Olga Polenogova, Elena Kosman, Darya Sidorenko, Tatyana Alikina, Yury Noskov, Anton Krivopalov, Viktor V. Glupov, Vadim Yu. Kryukov

**Affiliations:** 1Institute of Systematics and Ecology of Animals, Siberian Branch of Russian Academy of Sciences, Novosibirsk 630091, Russia; yarosl@inbox.ru (O.Y.);; 2Institute of Chemical Biology and Fundamental Medicine, Siberian Branch of Russian Academy of Sciences, Novosibirsk 630090, Russia; kabilov@niboch.nsc.ru (M.R.K.);

**Keywords:** soil microbiota, fungistasis, entomopathogenic fungus, *Beauveria*, *Metarhizium*, *Leptinotarsa decemlineata*, potato field, agricultural practice

## Abstract

Entomopathogenic fungi can be inhibited by different soil microorganisms, but the effect of a soil microbiota on fungal growth, survival, and infectivity toward insects is insufficiently understood. We investigated the level of fungistasis toward *Metarhizium robertsii* and *Beauveria bassiana* in soils of conventional potato fields and kitchen potato gardens. Agar diffusion methods, 16S rDNA metabarcoding, bacterial DNA quantification, and assays of *Leptinotarsa decemlineata* survival in soils inoculated with fungal conidia were used. Soils of kitchen gardens showed stronger fungistasis toward *M. robertsii* and *B. bassiana* and at the same time the highest density of the fungi compared to soils of conventional fields. The fungistasis level depended on the quantity of bacterial DNA and relative abundance of *Bacillus*, *Streptomyces*, and some *Proteobacteria*, whose abundance levels were the highest in kitchen garden soils. Cultivable isolates of bacilli exhibited antagonism to both fungi in vitro. Assays involving inoculation of nonsterile soils with *B. bassiana* conidia showed trends toward elevated mortality of *L. decemlineata* in highly fungistatic soils compared to low-fungistasis ones. Introduction of antagonistic bacilli into sterile soil did not significantly change infectivity of *B. bassiana* toward the insect. The results support the idea that entomopathogenic fungi can infect insects within a hypogean habitat despite high abundance and diversity of soil antagonistic bacteria.

## 1. Introduction

The soil microbiota includes bacteria, fungi, protozoa, viruses, and archaea, which interact with each other [1]. Soil microbial biomass is dominated by bacteria and fungi [2]. In 1953, Dobbs and Hinson [3] described the suppression of fungal-propagule germination by different soils, and this phenomenon was named as fungistasis. The intensity of fungistasis depends on many factors, including the soil’s physical and chemical properties, its microbial community, metabolic activity of the soil microbiota, and fungal characteristics [4]. In particular, the sensitivity to fungistasis differs among various ecological groups of fungi. For example, phytopathogenic fungi are more susceptible to fungistasis than saprotrophic fungi are [5]. Inhibition of fungal germination and mycelial growth is manifested by microorganisms belonging to different taxa and especially by bacteria. For instance, Zou et al. [6] studied the fungistatic activity of volatile compounds of soil bacteria toward nematodophillous fungi *Purpureocillium lilacinum* (formerly *Paecilomyces lilacinus*) and *Metacordyceps chlamydosporia* (formerly *Pochonia chlamydosporia*). Those authors showed that 328 bacterial isolates out of 1080—belonging to Alcaligenaceae, Bacillales, Micrococcaceae, Rhizobiaceae, and Xanthomonadaceae—exerted a fungistatic effect. Among them, 219 isolates inhibited the germination of spores and mycelial growth, whereas the activity of the remaining 109 isolates was limited only to the inhibition of mycelial growth.

Entomopathogenic ascomycetes *Metarhizium* and *Beauveria* are common in a variety of natural and agricultural habitats. These fungi can cause different levels of mortality in insect populations and are used as biocontrol agents against crop pests [7,8]. These pathogens infect hosts through integuments by means of mechanical pressure and a large set of hydrolytic enzymes [9]. Soil is a natural reservoir of these fungi, and their load usually is 10^2^–10^4^ colony-forming units (CFUs) per gram [10]. The insects killed by the fungi in most cases are located in the soil or on its surface. In addition to the parasitizing of insects, fungi *Beauveria* and *Metarhizium* are facultative plant symbionts. By interacting with the root system, they participate in exchanges of nitrogen, carbohydrates, phosphorus, and other elements with plants and cause growth-stimulating and immunomodulatory effects [11,12]. Successful persistence of entomopathogenic fungi in soil is influenced by a number of factors: temperature, pH, texture, organic matter content, water activity, and soil biotic components as reviewed by Jaronski [13]. Nevertheless, the relationship between the microbial community and insect pathogenic fungi is still insufficiently understood. An antagonistic impact of soil *Bacillus* and *Streptomyces* on entomopathogenic fungi has been demonstrated in several studies. For example, Popowska-Nowak et al. [14] found that bacteria *Bacillus subtilis* and *B. pumilus* have the greatest inhibitory effect on strains of *Cordyceps farinosa* (formerly *Paecilomyces farinosus*), *C. fumosorosea* (formerly *Paecilomyces fumosoroseus*), and *Beauveria bassiana* s.l. Additionally, volatile compounds of actinomycetes *Streptomyces flavescens* and *S. griseoviridis* completely inhibited the growth of all studied fungi; entomopathogenic fungi reacted differently to volatile and nonvolatile metabolites of the same soil microorganisms [14]. The inhibitory influence of soils of Western Siberia on fungi *B. bassiana*, *Metarhizium anisopliae* s.l., *C. farinosa*, and *C. fumosorosea* was shown by Sharapov and Kalvish [15]. In their study, the percentage of germinated conidia of entomopathogenic fungi was significantly lower in soddy-gley, forest podzolized, and leached chernozemic soils and depended on the season of soil sampling. In summer and autumn, these soils had the greatest inhibitory effect on the germination of conidia of all tested entomopathogenic fungi [15]. It can be speculated that the level of soil fungistasis influences the development of mycoses in insects and accordingly the effectiveness of mycoinsecticides. Nonetheless, the extent to which the incidence of insect fungal infections is related to soil fungal antagonists remains to be researched.

The type of agricultural practice can play an important role in the structure of communities of soil bacteria and fungi. For instance, the conventional tillage practice leads to a disturbance of the soil horizon and, as a result, to a decrease in biodiversity. The no-tillage practice, on the contrary, causes soil enrichment with organic matter and its restoration [16]. An investigation into microbiomes of fields with different agricultural practices is in progress [16,17,18,19,20]; however, the impact of the soil microbiota during different agricultural practices on fungistasis toward entomopathogenic fungi is not yet well characterized. 

The Colorado potato beetle (CPB) *Leptinotarsa decemlineata* (Say) is one of the most dangerous potato pests in the world. Having a monovoltine cycle, the CPB spends at least two periods of its life cycle in soil. The first period occurs during the autumn–winter–spring hibernation of adults, and the second one in the summer, when the larvae are buried into the soil for metamorphosis. With bi- and polyvoltine cycles, metamorphosis takes place several times during the summer. In both cases (metamorphosis and hibernation), beetles can be infected by fungi at the enzootic or epizootic level [21]. The most abundant entomopathogenic fungi in soils of potato fields in the temperate zone are *Metarhizium robertsii*, *M. brunneum*, and *B. bassiana* s.l. [22], and *B. bassiana* most often infects CPB larvae, pupae, and adults [23,24]. The influence of soils with different fungistasis levels on the infection of CPB pupae by entomopathogenic fungi was studied by Groden and Lockwood [25]. They showed a trend toward an increase in the 50% lethal dose (LD_50_) of *B. bassiana* conidia toward CPB pupae with an increase in soil fungistasis. Additionally, the level of fungistasis significantly affected the sporulation of *B. bassiana* on the cadavers of pupae. Nevertheless, the influence of the microbial community on these parameters was not examined.

The objective of the study was to discover possible relations between the level of fungistasis in soils of potato fields, the structure of the bacterial microbiota, and the development of fungal infections in the CPB during metamorphosis, in particular, (1) to examine the level of fungistasis and the CFU count of *Beauveria* and *Metarhizium* fungi in soils of potato fields subjected to different agricultural practices; (2) to analyze bacterial communities of the soils by metabarcoding sequencing of the 16S rRNA gene; (3) to isolate cultivable bacteria and assess their antagonistic activity against *M. robertsii* and *B. bassiana*; and (4) to evaluate the effect of soils with different fungistasis levels on the development of fungal infections in the CPB during metamorphosis as well as the impact of the introduction of antagonistic bacteria into the soil on the mortality of the CPB from mycoses.

## 2. Methods

### 2.1. Study Locations and Sampling

Soils of potato fields from three locations in Novosibirsk Oblast (Western Siberia) were analyzed: Karasuk (53°43′28″ N, 77°38′22″ E), Toguchin (55°02′46″ N, 84°49′09″ E), and Novosibirsk (55°03′39″ N, 82°45′30″ E). The selected locations are characterized by different agricultural practices and different textural and chemical composition of soils as described in detail previously [22]. Briefly, Karasuk fields are represented by sandy clay soils in kitchen gardens with long-term (more than 12 years) potato cultivation. Toguchin fields feature silty clay soils in kitchen gardens with continuous potato cultivation. Novosibirsk fields contain sandy clay loam soils in a conventional agrosystem with crop rotation and intensive farming using Dutch technology [26].

Samples of soil were randomly taken from each location in July 2021. Samples were collected from a depth of 5–15 cm at a distance of 20 m from each other. Each analyzed sample was a pool from three subsamples within a site having a radius of 4–5 m. The soils were placed in plastic bags and delivered to the laboratory within 24 h. 

### 2.2. Fungi and Insects

Isolates of entomopathogenic fungi *M. robertsii* (isolate P-72, GenBank # KP172147.2) and *B. bassiana* (isolate Sar-31, GenBank # MZ564259) from the collection of microorganisms at the Institute of Systematics and Ecology of Animals, the Siberian Branch of the Russian Academy of Sciences (SB RAS) were used. P-72 was isolated from the CPB in Latvia in 1972; Sar-31 was isolated from *Calliptamus italicus* in Karasuk district (Novosibirsk Oblast, Western Siberia) in 2001. The fungi were stored in a 10% aqueous glycerin solution at −80 °C and for experiments were cultivated on 1/4 Sabouraud-dextrose agar with 0.2% of yeast extract (SDAY) for 10 days at 26 °C in the dark.

Finishing feeding larvae of the CPB *L. decemlineata* (6–7 days postmolt in IV instar) served as test insects. The larvae were collected in Karasuk’s private potato fields that are free from application of biological insecticides. Larvae were maintained in the laboratory at 23 °C under a 16 h photoperiod and were fed with *Solanum tuberosum* plants.

### 2.3. Metarhizium and Beauveria CFU Counts in Soils

Five grams of each soil sample were added in 40 mL of an aqueous Tween-20 solution (0.1%) and shaken at 180 rpm for 1 h. Then, 100 μL of the soil suspension was plated in 90 mm Petri dishes containing the Sabouraud medium supplemented with antibiotics and fungistatics (glucose, 40 g/L; peptone, 10 g/L; yeast extract, 1 g/L; agar, 20 g/L; cetyltrimethylammonium bromide, 0.35 g/L; cycloheximide, 0.05 g/L; tetracycline, 0.05 g/L; and streptomycin, 0.6 g/L). The dishes were incubated at 25 °C for 14 d, and *Metarhizium* and *Beauveria* colonies were detected by light microscopy and counted. A weighed portion of each soil sample was dried to constant weight, and the CFU counts were normalized to the dry weight of the soils. Four biological replicates for each location were analyzed.

### 2.4. Fungistatic Activity of Soils toward M. robertsii and B. bassiana

Soil aqueous suspensions were prepared as follows: 5 g of soil with 40 mL of sterile distilled water were shaken at 120 rpm for 1 h. Entomopathogenic fungi *M. robertsii* (strain P-72) and *B. bassiana* (strain Sar-31) were inoculated and spread with a spatula on SDAY in 90 mm Petri dishes. After that, a soil extract (a 5 μL aliquot) was placed dropwise into 2 mm wells in the medium. The Petri dishes were incubated at 25 °C, and growth inhibition zones were measured on the 4th day. The experiment was conducted on 4 biological replicates and 3 technical replicates of soil from each location.

### 2.5. 16S rDNA Metabarcoding

One gram of each soil sample was placed into 15 mL tubes, frozen in liquid nitrogen, and stored at −80 °C until analysis. Four biological replicates for each location were analyzed. Total DNA was extracted from 100–200 mg of soil using the DNeasy PowerSoil Pro DNA Isolation Kit (Qiagen, Hilden, Germany) as per the manufacturer’s instructions. The bead-beating was performed by means of TissueLyser II (Qiagen, Hilden, Germany) for 10 min at 30 Hz.

The V3–V4 region of the 16S rRNA gene was amplified with primer pair 343F (5′-CTCCTACGGRRSGCAGCAG-3′) and 806R (5′-GGACTACNVGGGTWTCTAAT-3′) combined with Illumina adapter sequences [27]. PCR amplification was performed as described earlier [19]. A total of 200 ng of a PCR product from each sample was prepared (by combining three technical replicates) and purified with the MinElute Gel Extraction Kit (Qiagen). The 16S libraries were sequenced with 2 × 300 bp paired-end reagents on MiSeq (Illumina, CA, USA) at the SB RAS Genomics Core Facility (ICBFM SB RAS, Novosibirsk, Russia).

Raw sequences were analyzed via the UPARSE pipeline [28] using Usearch v11.0.667. The UPARSE pipeline included the merging of paired reads, read quality filtering, length trimming, merging of identical reads (dereplication), discarding singleton reads, removal of chimeras, and operational taxonomic unit (OTU) clustering by the UPARSE-OTU algorithm [29]. The OTU sequences were assigned to taxa using SINTAX [30,31] and 16S RDP training set v18 (from 2021.02) as a reference [32]. Alpha diversity metrics were calculated in Usearch. Rarefaction and extrapolated curves were generated with the help of the iNEXT package [33].

The final dataset consisted of 793,959 (66,163 ± 3059 per sample) reeds affiliated with 6866 OTUs (see Appendix A). All rarefaction curves tended to reach a plateau (Appendix A), indicating sufficient sequencing depth.

### 2.6. Bacterial DNA Quantification

This quantification was performed by real-time PCR on a CFX96 Touch real-time PCR detection system (Bio-Rad, Hercules, CA, USA). Primers 1369F (5′-CGGTGAATACGTTCYCGG-3′) and 1492R2 (5′-GGWTACCTTGTTACGACTT-3′) specific to the V9 region of the 16S rRNA gene were employed [34]. PCR amplification was performed in 20 μL reaction mixtures composed of 0.28 U of Phusion Hot Start II High-Fidelity and 1× Phusion GC buffer (Thermo Fisher Scientific, Waltham, MA, USA), 0.5 μM each forward and reverse primers, 2.3 mM MgCl_2_ (Sigma-Aldrich, St. Louis, MO, USA), and 0.2 mM each dNTP (Life Technologies, Carlsbad, CA, USA). Thermal cycling conditions were as follows: initial denaturation at 98 °C for 30 s followed by 40 cycles of 98 °C for 10 s, 55 °C for 10 s, and 72 °C for 15 s with final extension at 72 °C for 5 min. Standard curves for bacteria were generated by quantitative PCR analysis of serial 10-fold dilutions of bacterial DNA standards (Zymo Research, E2006-2, Irvine, CA, USA,).

### 2.7. Isolation and Identification of Soil Bacteria

Soil aqueous suspensions were prepared as described above (Section 2.3). The suspensions were diluted by 10^−2^, 10^−3^, and 10^−4^. Aliquots (100 μL) were inoculated onto Luria–Bertani agar (LB) in 90 mm Petri dishes and incubated for 5 days at 28 °C. Representative colonies were isolated and subcultured three times for purification and subsequent identification. 

For DNA extraction, each bacterial colony was stirred in 1 mL of autoclaved distilled water (ddH_2_O) and centrifuged for 1 min at 11,000 rcf; the supernatant was gently removed. Next, 200 µL of a 5% aqueous suspension of Chelex 100 Resin (Bio-Rad Laboratories, Inc., USA) was added to the precipitate and incubated for 30 min at 56 °C. The samples were vortexed for 10 s, and the tubes were heated at 100 °C for 8 min. After that, the samples were vortexed again for 10 s and centrifuged for 3 min at 15,400 rcf. Finally, 20 µL of the resulting supernatant was subjected to PCR.

The PCR primers were 27F (5′-AGA GTT TGA TCA TGG CTC AG-3′) [35] and 1492R (5′-CCC TAC GGT TAC CTT GTT AGG ACT-3′), which allow all variant regions of the 16S rRNA gene to be covered. The PCR was carried out in 50 µL reaction mixtures composed of 5× PCR buffer from Biolabmix (Novosibirsk, Russia) (50 mM Tris-HCl, 250 mM KCl, 7.5 mM MgCl_2_, 0.5% [*v*/*v*] of Tween-20), 0.2 mM each dNTP, 0.25 µM each primer, and 1.5 units of Hot Star Taq DNA polymerase (Biolabmix). The thermal cycling program consisted of an initial denaturation step (95 °C for 5 min) followed by 29 cycles of 95 °C for 10 s, 60 °C for 15 s, and 72 °C for 60 s with a final extension at 72 °C for 7 min and a holding temperature of 12°C.

The obtained amplicons were visually evaluated in a 1.5% agarose gel after electrophoresis in 1× TAE buffer. PCR products were purified and sequenced on an ABI 3130xl genetic analyzer (Applied Biosystems, Foster City, CA, USA) at the SB RAS Genomics Core Facility (ICBFM SB RAS, Novosibirsk, Russia) using Big Dye 3.1 chemistry (Thermo Fisher Scientific, Vilnius, Lithuania). The sequences were edited and assembled by means of FinchTV 1.5 software (Geospiza Inc., Seattle, WA, USA). The similarity of the original nucleotide sequences to homologs was assessed on the NCBI BLAST web site [36] with the BLASTn program.

### 2.8. Antagonism of Soil Bacteria against Entomopathogenic Fungi 

This parameter was analyzed by an agar plug diffusion method [37]. Briefly, plugs of 1-day-old bacterial cultures (in LB) were placed on a freshly plated culture of *M. robertsii* or *B. bassiana* in 90 mm Petri dishes containing SDAY. The Petri dishes were incubated at 26 °C in the dark. The inhibition zone of mycelial growth (mm) was evaluated during 4 days of incubation. Four biological replicates for each bacterium were tested in the assay.

### 2.9. Fungal Infection of the CPB in Soils with Different Fungistasis Levels

Unsterilized soils from three locations (Karasuk, Toguchin, and Novosibirsk) were used in this experiment, which was conducted under two conditions: (1) with natural initial soil moisture and (2) in soils with standardized matric potential (equalized among all analyzed samples). In the first experiment, matric potential was measured with Irrometer SR-12 (Irrometer Co., Riverside, CA, USA) and proved to be −22 kPa in Karasuk, −36 kPa in Toguchin, and −14 kPa in Novosibirsk soils. Next, 200 g soil samples were placed in 1000 mL plastic containers. Dry conidia of *B. bassiana* Sar-31 were added to the soil samples to obtain a final concentration of 10^4^, 10^5^, and 10^6^ conidia/g, and the samples were thoroughly homogenized. Control samples were processed analogously but without the fungal treatment. Finishing feeding larvae were placed on the soil surface (10 larvae per container) for self-guided burial. Containers were incubated at 21 °C under the 16 h photoperiod. Mortality was registered after 30 days of incubation, which corresponded to the period from deposition of larvae in soil to emergence of adults. Five biological replicates were used for each treatment group (one replicate = 10 larvae). 

In the second experiment, matric potential of studied soils was equalized to −30 kPa by drying at room temperature or by the gradual addition of sterile water and subsequent measurement with Irrometer SR-12. Dry conidia of *B. bassiana* Sar-31 were introduced to obtain final concentration 2 × 10^6^ conidia/(g of soil). Control samples were not subjected to the fungal treatment. Introduction of beetles and incubation of the containers were conducted as described above. Four biological replicates were utilized for the control and six biological replicates for fungal treatment (one replicate = 10 larvae). 

### 2.10. The Influence of Soil Bacteria on Fungal Infection in the CPB

Two bacterial isolates (*Bacillus frigoritolerans* and *B. pumilus*) having high antagonistic activity toward fungi as well as the fungus *B. bassiana* were chosen for the assay. Soil from the Karasuk location was autoclaved for 1 h under 1.2 Atm two times with a 1-day interval. Then, each soil sample was dried at 160 °C for 3 h. Soil samples were placed in 1000 mL plastic containers (200 g of soil per container), and 13 mL of sterile water (as determined on the basis of water evaporation compared to nonautoclaved soil) was added to each sample. One milliliter of a *B. frigoritolerans* or *B. pumilus* suspension was added to each sample to obtain a final concentration of 10^7^ cells/(g of soil). Sterile water was added to control samples. Then, the soil samples were inoculated with 1 mL of a *B. bassiana* aqueous Tween-20 (0.03%) suspension or a conidia-free aqueous Tween-20 solution to attain final concentrations of 0, 5 × 10^6^, 2 × 10^6^, and 5 × 10^6^ conidia per gram of soil. Soil samples were thoroughly homogenized. Ten finishing feeding larvae and one potato leaf were put on the surface of the soil in each container. The larvae buried themselves into the soil during 1–2 days. The containers were incubated as described above. Mortality levels were assessed on day 30 of incubation (period of metamorphosis). Six biological replicates were analyzed in each treatment and control group (one replicate = 10 beetles).

### 2.11. Statistics

Data analysis was performed using STATISTICA 8 (StatSoft Inc., Tulsa, OK, USA) and PAST 4.03 [38]. The normality of the data distribution was checked by the Shapiro–Wilk *W* test. Normally distributed data were subjected to one-way or two-way ANOVA (depending on experimental setup) followed by Tukey’s post hoc test. Non-normally distributed data were subjected to Kruskal–Wallis ANOVA or a nonparametric equivalent of two-way ANOVA Scheirer–Ray–Hare test [39] followed by Dunn’s post hoc test. Data in plots and tables are presented as an arithmetic mean and standard error.

## 3. Results

### 3.1. Fungistatic Activity of Soil Extracts and Fungal CFU Counts

Two-way ANOVA showed a significant inhibitory effect of soil extracts on fungal growth (effect of soil: H_2,23_ = 15.8, *p* = 0.0004, Figure 1A); however, there were no differences in the level of inhibition of *M. robertsii* or *B. bassiana* (effect of a fungus: H_1,23_ = 0.1, *p* = 0.77). The growth of both entomopathogenic fungi was most strongly suppressed by soil extracts from Karasuk kitchen gardens. Minimal suppression was manifested by soil extracts from Novosibirsk conventional fields (Dunn’s test *p* < 0.005 as compared to Karasuk soil). Soil extracts from Toguchin kitchen gardens showed an intermediate level of fungal growth inhibition and the absence of significant differences from Karasuk and Novosibirsk soil extracts (*p* > 0.08). 

CFU counts of *Metarhizium* and *Beauveria* in the soils had patterns similar to those of fungistatic activity (Figure 1B). The CFU count was significantly higher in Karasuk and Toguchin kitchen garden soils than in soils from Novosibirsk conventional fields (Dunn’s test *p* < 0.04). The correlation between the CFU count and fungistatic activity was positive, but significance was marginal (r = 0.78, *p* = 0.07). Thus, highly fungistatic soils are characterized by the highest fungal CFU count.

### 3.2. 16S rDNA Metabarcoding Analysis

Among all samples, 6866 OTUs belonging to 32 phyla, 87 classes, 138 orders, 265 families, and 494 genera were detected (Appendix A). The communities showed predominance of phyla Acidobacteria (relative abundance of 32%), Actinobacteria (31%), Proteobacteria (14%), and Firmicutes (6%). Phyla Gemmatimonadetes, Chloroflexi, Verrucomicrobia, and Bacteroidetes had relative abundance of 1–3%. 

The microbial communities from studied locations were similar in the composition of phyla relative abundance but differed in distribution of lower taxa (Figure 2A–C). Soils from kitchen gardens (Karasuk, Toguchin) had 1.2–1.4-fold lower abundance of Acidobacteria as compared to soils from conventional fields (Novosibirsk) (Tukey’s test, *p* < 0.01). Relative abundance of Actinobacteria was similar among all soils: 29–33% (*p* > 0.07, Figure 2A), but relative abundance of various genera within this taxon differed among the soils (Figure 2C). In highly fungistatic soils (Karasuk), relative abundance of *Streptomyces*, unclassified (unc.) Solirubrobacterales, and unc. Micromonosporaceae was rather high (*p* < 0.001, *p* < 0.03, and *p* < 0.001, respectively, relative to other soils). In soil with intermediate fungistasis (Toguchin), an increased relative abundance of actinobacteria *Gaiella* and *Microlunatus* was documented (*p* < 0.001 relative to other soils). In weakly fungistatic soil from Novosibirsk conventional fields, unc. Actinobacteria and *Pseudarthrobacter* were predominant (*p* < 0.003 relative to other soils).

Soils from Karasuk and Toguchin kitchen gardens had 1.3–1.5-fold higher relative abundance of Proteobacteria than did soil from Novosibirsk conventional fields (*p* < 0.004, Figure 2A). This result was mainly due to elevated Alphaproteobacteria relative abundance (Figure 2B), in particular *Sphingomonas*, whose proportion was the highest in Karasuk soils (*p* < 0.002 relative to other soils), unc. Rhizobiales predominant in Toguchin soils (*p* < 0.002 relative to other soils), and *Bradyrhizobium*, which was present only in Karasuk and Toguchin soils (Figure 2C).

Significant differences were detected among the communities in relative abundance of Firmicutes, in particular, Bacilli (Figure 2A,B). In high-fungistasis soils from kitchen gardens (Karasuk and Toguchin), a 3.5- to 3.8-fold higher relative abundance of Firmicutes and Bacilli was registered relative to low-fungistasis soils from Novosibirsk conventional fields (*p* < 0.01). Predominant genera were *Bacillus* and *Paenibacillus*, whose relative abundance levels were the lowest in Novosibirsk soils (*p* < 0.01 in comparison with other soils, Figure 2C).

Indices that are sensitive to the dominance of taxa (Simpson 1-d and Berger–Parker 1/d) had the lowest value in soil communities from Novosibirsk conventional fields (Dunn’s test, *p* < 0.03 relative to other locations, Figure 2E,F); however, Shannon index values were similar among all soils, with minor but significant elevation at the Toguchin location (Tukey’s test *p* < 0.001, Figure 2G). The index of richness Chao1 was the highest in Toguchin soil (Dunn’s test, *p* < 0.05), although its values did not significantly differ between Karasuk and Novosibirsk soils (*p* = 0.4, Figure 2D). 

Thus, soils from kitchen gardens and a high level of fungistasis had elevated relative abundance of Bacilli and Alphaproteobacteria and a specific structure of Actinobacteria communities, in particular, a higher portion of *Streptomyces*. In soils with low fungistasis, diminished values of dominance indices (Simpson 1-d and Berger–Parker 1/d) were registered; however, the richness index (Chao1) did not correlate with the fungistasis level. 

### 3.3. Total Bacterial Load 

Soils from kitchen gardens (Karasuk and Toguchin) had a dramatically larger (25–83-fold) total amount of bacterial DNA as compared to soils from Novosibirsk conventional fields (Tukey’s test, *p* < 0.003, Table 1). Notably, soils with intermediate fungistasis (Toguchin) possessed a 3.4-fold greater total amount of bacterial DNA than did soils with the highest fungistasis (Karasuk) (*p* < 0.001). The pattern was similar for the ranking of the soils by Simpson 1-d and Berger–Parker 1/d values. 

### 3.4. The Influence of Cultivable Bacteria on Fungi

We isolated 22 cultures of soil bacteria using the LB medium (Table 2). Most of them were affiliated with Bacilli (genera *Bacillus*, *Brevibacillus*, *Peribacillus*, and *Psychrobacillus*) and only two cultures belonged to Actinobacteria (*Arthrobacter bambusae*) and Betaproteobacteria (*Janthinobacterium lividum*). Because soils with different fungistasis levels differed in Bacilli relative abundance, we analyzed antagonistic properties of these bacteria toward *B. bassiana* and *M. robertsii.* Another purpose of this work was the selection of representative isolates for a bioassay on the CPB. 

Various bacteria showed different inhibitory effects on fungal growth (H_21,175_ = 117, *p* < 0.001, Figure 3), but there were no significant differences in suppression of *B. bassiana* or *M. robertsii* growth (H_1,175_ = 0.01, *p* = 0.93). Isolates of *Bacillus frigoritolerans*, *B. mobilis*, and *B. subtilis* showed a strong fungistatic activity against both fungi. Among *B. pumilus* isolates, there were those that actively inhibited the growth of the fungi and isolates that did not exhibit the antagonistic activity. Moderate antagonistic activity was manifested by *Psychrobacillus psychrodurans* and *Brevibacillus formosus*. Isolates of *Bacillus thuringiensis*, *Peribacillus simplex*, *Brevibacillus laterosporus*, *A. bambusae,* and *J. lividum* showed little or no antagonistic action toward both fungi. 

### 3.5. Effects of Different Soils on Fungal Infection in the CPB

For this assay, we chose *B. bassiana* as a fungal pathogen more often killing the CPB in soil. Experiments were performed on nonsterile soils with different fungistasis levels and having (1) an unaltered moisture content or (2) matric potential equalized to −30 kPa. In the case of unaltered moisture, there were no significant effects on mortality depending on the soil type (H_2,29_ < 1.7, *p* > 0.19, Figure 4A). Nonetheless, trends of decreased mortality in weakly fungistatic soils (Novosibirsk conventional fields) were observed. In particular, at an intermediate concentration of conidia (10^5^/[g of soil]), mortality was 4-fold higher in highly fungistatic soil (Karasuk) than in low-fungistasis soil from Novosibirsk (Dunn’s test, *p* = 0.03) although Karasuk soil was drier (−22 kPa) than Novosibirsk soil (−14 kPa). A similar pattern was observed at this concentration between the soil with intermediate (Toguchin) and low (Novosibirsk) fungistasis, but differences were not significant (*p* = 0.12). 

In the case of matric potential equalized among the studied soils to −30 kPa, the highest mortality was registered in highly fungistatic soil (Karasuk): 1.4–1.7-fold greater than mortality in soils with intermediate and low fungistasis, from Toguchin and Novosibirsk, respectively (Dunn’s test, *p* < 0.03, Figure 4B). Notably, mortality in untreated controls was observed only in Karasuk soil and amounted to 26%. All cadavers in the abovementioned experiments (including cadavers from controls) exhibited symptoms of mycoses, i.e., were overgrown by the *Beauveria* mycelium. 

### 3.6. The Effect of Antagonistic Bacteria on Fungal Infection in the CPB in Sterile Soil

For this assay, we chose two isolates of *B. frigoritolerans* and *B. pumilus* exhibiting strong antagonistic action toward the fungi. Effects of inoculation of sterile soil with bacteria and fungal conidia on the mortality of the CPB were evaluated. 

The introduction of bacteria alone into sterile soil did not cause an increase in mortality in comparison with the control (Figure 5A,B). Joint introduction of bacteria and a fungus did not lead to significant differences in the mortality level either, when compared to soil inoculation with a fungus alone (the effect of *B. pumilus*: F_1,40_ = 0.04, *p* = 0.84; the effect of *B. frigoritolerans*: F_1,40_ = 2.34, *p* = 0.13). Nevertheless, at the lowest concentration of conidia (5 × 10^5^), we noticed a slight rise of mortality when *B. bassiana* and *B. frigoritolerans* were combined (Tukey’s test, *p* = 0.07, as compared to treatment with a fungus alone, Figure 5B). All cadavers exhibited symptoms of mycoses. In other words, *B. frigoritolerans* raised fungus-induced mortality via an additive effect. 

## 4. Discussion

Herein, using potato field soils, we show relations among the level of fungistasis toward entomopathogenic fungi, the bacterial communities, and the infectivity of *B. bassiana* toward the CPB during metamorphosis. Antagonistic effects on entomopathogenic fungi were more pronounced in kitchen garden soils (Karasuk) than in soils of conventional fields (Novosibirsk), in agreement with the high quantity of total bacterial DNA and high relative abundance of Firmicutes (Bacilli), Proteobacteria, and certain taxa of Actinobacteria (*Streptomyces*) in kitchen garden soils. Nevertheless, soils with higher fungistasis contained higher CFU counts of *Beauveria* and *Metarhizium*. Moreover, we registered trends of elevated infectivity of *B. bassiana* toward CPB in highly fungistatic soils compared to low-fungistasis ones.

In soils of conventional fields, in addition to the low quantity of total bacterial DNA, we registered the lowest Simpson 1-d and Berger–Parker 1/d indices as compared with the kitchen garden soils. This finding is obviously due to greater soil degradation under intensive farming. It is well known that conventional field soils are characterized by reduced diversity and density of bacteria in comparison with organic field soils. For example, Kraut-Cohen et al. [16] demonstrated that Israel farms not subjected to tillage have the highest bacterial diversity as compared to agricultural systems with conventional tillage and minimal tillage. Those authors showed that even minimal tillage leads to a change in the microbial-community structure. Soil analysis in corn fields of conventional and organic farming systems in the Netherlands revealed that the microbiota are more heterogeneous in organic than in conventional farming systems [40]. Our results confirm that conventional farming systems are characterized by decreased microbial density and diversity.

We found that one of the main distinctive features of bacterial composition of soils with high fungistasis (kitchen gardens) is an elevated relative abundance of Bacilli, specifically *Bacillus* and *Paenibacillus*. There is evidence of antagonistic properties of *Bacillus* species residing in the soil and rhizosphere (e.g., *B. pumilus*, *B. subtilis*, *B. cereus*, and *B. amyloliquefaciens*) toward phytopathogenic fungi, such as *Botrytis*, *Alternaria*, *Fusarium*, *Colletotrichum*, *Doratomyces*, *Penicillium*, *Pyricularia*, and *Sclerotinia* [41,42,43,44]. Antagonistic action on *Beauveria* or *Metarhizium* species has been documented in vitro for soil- and insect-associated *B. pumilus* [14,45,46], *B. subtilis* [14,45,47], *B. solani* [48], *Bacillus thuringiensis*, *B. mycoides*, *B. megaterium*, and *B. licheniformis* [45]. Such bacilli as *Bacillus* and *Paenibacillus* produce a set of volatiles, e.g., benzaldehyde, benzothiazole, dimethyldisulfide, 1-undecene, 1-octen-3-ol, and citronellol, that suppress filamentous fungi [49]. Additionally, these bacteria produce a number of antifungal peptides and macrolides, such as iturin A, bacillomycin L, bacillopeptins, rhizocticin A, fengycin, and fungicidin [50]. We also demonstrated that bacilli from potato field soils exert antagonism toward both *B. bassiana* and *M. robertsii*, wherein *B. frigoritolerans*, *B. mobilis*, and some strains of *B. pumilus* and *B. subtilis* had the strongest effect.

Apparently, another important factor responsible for the differences in fungistasis among the studied soils is bacteria from the genus *Streptomyces*. Highly fungistatic soils (Karasuk location) were found to have the highest relative abundance of these bacteria. It is known that *Streptomyces* spp. have an antagonistic effect on fungi of various taxa. These actinobacteria are producers of a large set of volatiles, among which there are fungal inhibitors 2-phenylethanol, phellandrene, benzaldehyde, and dimethyldisulfide [49]. In addition, *Streptomyces* spp. produce many antifungal factors, including amphotericin, nystatin, phoslactomycins, kanchanamycins, phthoxazolins, dorrigocins, faeriefungin, polyenes, stendomycin, sultriecin, dunaimycins, phosmidosine, polyoxin, and others [50]. There are many examples of *Streptomyces* species being able to suppress entomopathogenic fungi in in vitro and in vivo systems. Volatiles of *Streptomyces griseoviridis* and *S. flavescens* cause complete inhibition of the growth of *B. bassiana*, *C. farinosa*, and *C. fumosorosea* [14]. Inoculation of *S. griseoviridis* into soil in laboratory experiments reduces the CFU count of *B. bassiana* [51]. In the cuticle, “leaf-cutter” ants have a profile of *Streptomyces* spp. that reduces infection by *M. anisopliae* [52] and suppresses the mycoparasite *Escovopsis*, which attacks ant fungal gardens [53]. *Streptomyces* from feces of the subterranean termite *Coptotermes formosanus* in its nest structure increase survival after *M. anisopliae* infection [54]. Li et al. [55] reported that *Streptomyces* associated with the termite *Macrotermes barneyi* produce polyenes possessing higher activity against *Xylaria* sp. and *M. anisopliae*. Disruption of polyene-related genes attenuated the antagonistic actions against both fungi.

In our work, highly fungistatic soils had higher relative abundance of Alphaproteobacteria, in particular, unc. Rhizobiales, *Bradyrhizobium*, and *Sphingomonas*. Rhizobiales and, in particular, *Bradyrhizobium* are known to produce various antifungal metabolites active against soil-born mycelial fungi, such as *Fusarium*, *Rhizoctonia*, and *Macrophomina* [56], although studies on entomopathogenic fungi are not known. *Sphingomonas paucimobilis* (formerly known as *Pseudomonas paucimobilis*) has been shown to exert antagonism toward the phytopathogenic fungus *Verticillium dahliae* [57], but research on entomopathogenic fungi is also lacking. In our study, many taxa of bacteria also significantly differed in relative abundance among soils having different fungistasis levels, for example, unc. Solirubrobacterales, unc. Gemmatimonadaceae, and *Gaiella*. Unfortunately, there are almost no data on interactions with fungi. This topic may become a subject of future research.

Our analysis of the CPB mortality rate in soils with the introduction of *B. bassiana* conidia uncovered trends toward increased mortality in highly fungistatic soils (Karasuk) compared to weakly fungistatic soils (Novosibirsk). This pattern was observed at equalized matric potential in soils (Figure 4B). Furthermore, this pattern was registered at certain concentrations of the fungus (10^5^ conidia/g), even if the high-fungistasis soils were less moist than low-fungistasis ones (Figure 4A). In contrast, Jaronski [13] showed that mortality of the Southern corn rootworm *Diabrotica undecimpunctata* is higher in sterile soil inoculated with *B. bassiana* than in nonsterile soil treated with this fungus. Likewise, Groden and Lockwood [25] documented a tendency of lower mortality of CPB pupae from *B. bassiana* with an increase in soil fungistasis. By contrast, Borisov [58] showed that a decrease in soil fungistasis via the introduction of various inducers of conidial germination (chitin, glucose, peptone, and yeast extract) sharply reduces the infection of the beetle *Pyrrhalta viburni* with the fungus *B. bassiana* under field conditions. In laboratory assays, the introduction of these germination inducers into soils weakened the infectivity of *Isaria*, *Metarhizium*, and *Lecanicillium* against various coleopteran species and wax moth *Galleria mellonella*. Quintela and coworkers [59] noticed that the survival rate of *B. bassiana* conidia is higher in nonautoclaved soil than in autoclaved soil. Apparently, the discrepancies in the effects may be due to developmental stages of an insect, the duration of exposure, and physical and chemical features of soils. We assume that in our experiments, higher soil fungistasis allows conidia to remain viable longer and to be activated precisely at the time of contact with the cuticle of a living host. Nevertheless, additional assays are needed to prove this hypothesis. It is noteworthy that the introduction of soil extracts into agarized media for entomopathogenic fungi suppresses mycelial growth but strongly raises conidial yield [58]. This report is also consistent with the higher CFU counts of *Metarhizium* and *Beauveria* that we observed in soils having high fungistasis (Figure 1). 

After the joint introduction of *B. bassiana* and its antagonists *B. frigoritolerans* and *B. pumilus*, there were no significant effects on the CPB mortality. Nevertheless, at low concentrations of conidia, the combination of *B. bassiana* and *B. frigoritolerans* increased mortality as an additive effect when compared with the inoculation with *B. bassiana* alone (*p* = 0.07). We believe that this outcome is due to (1) an enhancement of fungistasis, and as a result, longer survival of *B. bassiana* conidia; (2) certain pathogenicity of *B. frigoritolerans* to the CPB during the development of mycoses. It is known that conidia of *B. bassiana* can germinate in sterile but not in nonsterile soils [13], while fungistasis of sterile soils can be restored by the introduction of soil bacteria [60]. Therefore, the introduction of bacteria may contribute to higher infectivity of *B. bassiana*. Selvakumar et al. [61] documented the entomopathogenic properties of *B. frigoritolerans* toward *Holotrichia longipennis* and *Anomala dimidiata* (Coleoptera, Scarabaeidae). Oral bacterial infection of CPB larvae, which bury themselves into soil, was unlikely in our experiments. Nonetheless, during development of mycosis in the CPB, bacteria can also penetrate into the hemocoel through injuries in the cuticle [62]. Bacteria with fungi can have a synergistic or additive effect on insect mortality, even if the microorganisms demonstrate antagonism in vitro. The latter was reported, for example, by Augustyniuk-Kram et al. [51] about *S. griseoviridis*, *B. bassiana*, and *G. mellonella*, by Lednev et al. [63] on *Pseudomonas* sp., *B. bassiana*, and *Locusta migratoria*, and by Noskov et al. [64] on *Pseudomonas putida*, *M. robertsii*, and *Aedes aegypti*. Nonetheless, additional experiments are required to clarify the exact causes of increasing CPB mortality under the combined action of *B. bassiana* and *B. frigoritolerans* in soil.

## 5. Conclusions

This paper shows a difference in fungistasis among various soils (from potato fields) toward entomopathogenic fungi *B. bassiana* and *M. robertsii*. The kitchen garden soils proved to be more fungistatic to these fungi as compared to conventional field soils. The level of fungistasis depended on the quantity of total bacterial DNA and relative abundance of Bacilli, *Streptomyces*, and some Alphaproteobacteria. Bacilli isolated from the soil of potato fields showed antagonism toward both *B. bassiana* and *M. robertsii*. Nevertheless, *Beauveria* and *Metarhizium* CFU counts in soils positively correlated with levels of fungistasis. Moreover, trends toward higher mortality of the CPB from *B. bassiana* were observed in highly fungistatic soils as compared to weakly fungistatic ones. Both effects may indicate better survival of conidia in highly fungistatic soils. These results are in line with the general notion that *Beauveria* and *Metarhizium* germinate in the soil mainly in the presence of stimuli (insect cuticle), and these fungi are able to infect hosts within the hypogean habitat despite high density and diversity of soil antagonistic microbes [13,65]. Further research can be aimed at clarifying the impact of the fungal microbiota on fungistatic properties of soils toward entomopathogenic fungi and on the development of mycosis in insects in soils.

## Figures and Tables

**Figure 1 microorganisms-11-00943-f001:**
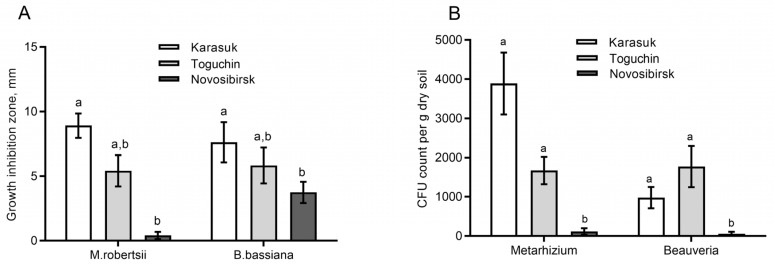
Antagonistic effects of soil extracts from soils of kitchen potato gardens (Karasuk and Toguchin) and from a conventional potato field (Novosibirsk) on fungi *M. robertsii* and *B. bassiana* (**A**) and on fungal CFU counts in these soils (**B**). Different letters indicate significant differences between soils for each fungus (Dunn’s test *p* < 0.05).

**Figure 2 microorganisms-11-00943-f002:**
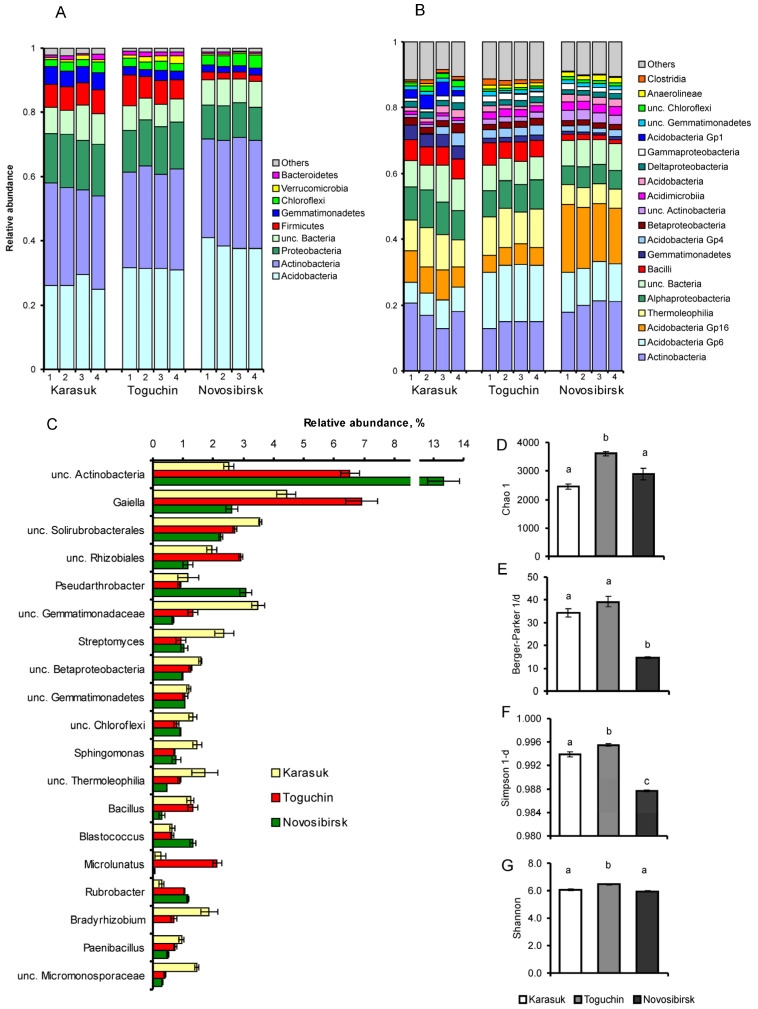
A 16S metabarcoding analysis of bacterial communities in the soils of potato fields with different levels of fungistasis and different agricultural practices: Karasuk (kitchen gardens, high level of fungistasis), Toguchin (kitchen gardens, intermediate level of fungistasis), and Novosibirsk (conventional fields, low fungistasis). Four samples for each soil are presented: (**A**) the distribution at the level of phyla; (**B**) the distribution at the class level; (**C**) the distribution at the level of genera; and (**D**–**G**) diversity indices. Different letters denote significant differences (Tukey’s test or Dunn’s test, *p* < 0.05).

**Figure 3 microorganisms-11-00943-f003:**
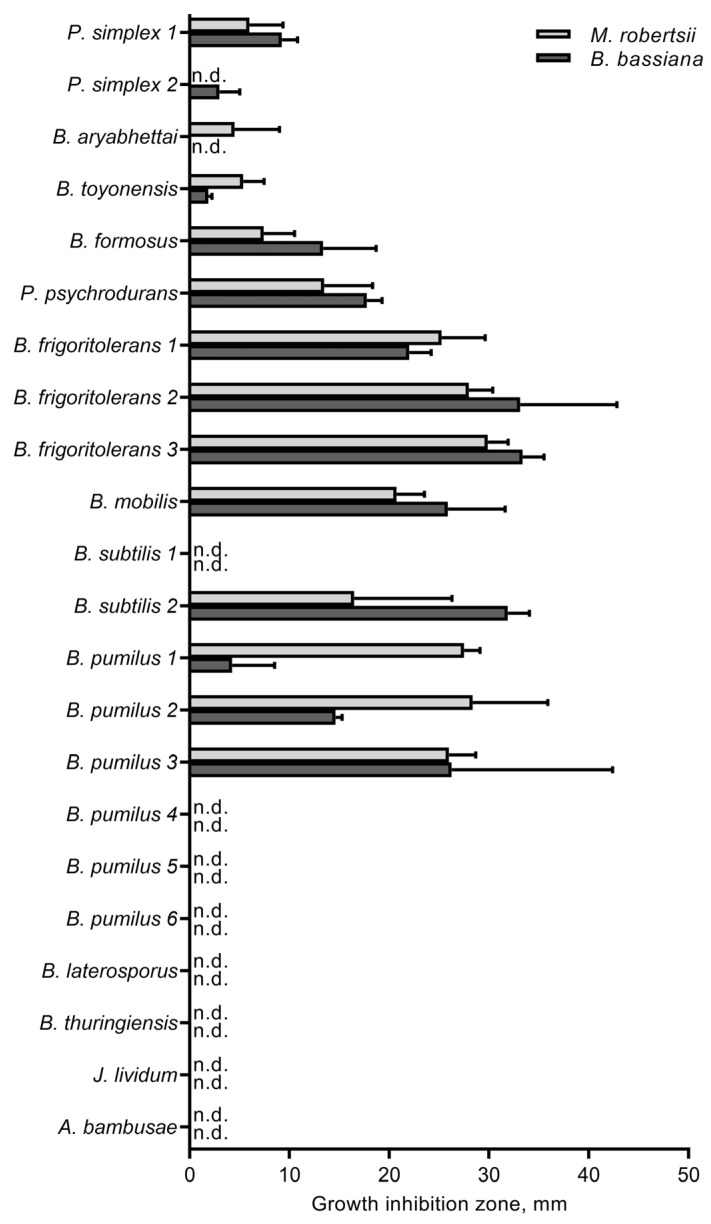
Antagonistic action of soil bacteria against entomopathogenic fungi *M. robertsii* and *B. bassiana,* as estimated by the agar plug diffusion method. n.d.: an inhibition zone was not detectable.

**Figure 4 microorganisms-11-00943-f004:**
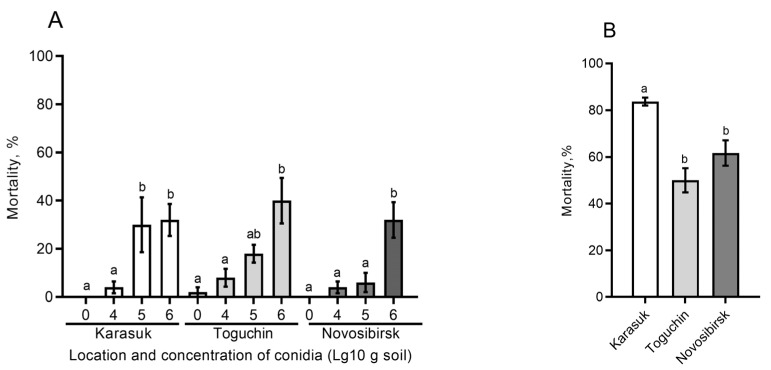
Mortality of the CPB during 30 days after burial into nonsterile soils having natural moisture and inoculated with *B. bassiana* conidia at three concentrations (**A**), and the same parameter for soils with matric potential equalized to −30 kPa and inoculated with *B. bassiana* conidia at a concentration of 2 × 10^6^ conidia per g (**B**); mortality was normalized to the control via the Abbott formula in the assay with equalized matric potential (**B**). Different letters indicate significant differences between treatment groups (Dunn’s test, *p* < 0.05).

**Figure 5 microorganisms-11-00943-f005:**
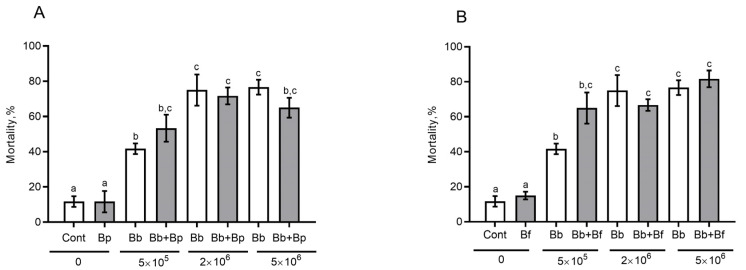
CPB mortality during 30 days after burial into the sterile soil inoculated with *B. bassiana* (Bb) conidia at three concentrations and with soil bacteria at the concentration 10^7^ cells/(g of soil): (**A**) *Bacillus pumilus* (Bp); (**B**) *B. frigoritolerans* (Bf). Different letters indicate significant differences between treatment groups (Tukey’s test *p* < 0.05).

**Table 1 microorganisms-11-00943-t001:** Total amounts of bacterial DNA in soils of potato fields with different levels of fungistasis and different agricultural practices: Karasuk (kitchen gardens, high fungistasis), Toguchin (kitchen gardens, intermediate level of fungistasis), and Novosibirsk (conventional fields, low fungistasis).

Location	Total Amount of Bacterial DNA, ng/mg	Tukey’s Test		
		Karasuk	Toguchin	Novosibirsk
Karasuk	5.7 ± 0.48	-	0.0001	0.003
Toguchin	19.22 ± 1.3	0.0001	-	0.0001
Novosibirsk	0.23 ± 0.09	0.003	0.0001	-

**Table 2 microorganisms-11-00943-t002:** Putative identification of bacteria (isolated from the soils) by means of 16S rRNA (~1390 bp) gene sequences.

Isolate and Location of Soil	Nearest Isolate from GenBank	Accession Number in GenBank	Identity, (%)
30720(T)	*Peribacillus simplex* NBRC 15720 = DSM 1321	NR_042136	99.77
30920(N)	*Peribacillus simplex* NBRC 15720 = DSM 1321	NR_042136	99.65
30220(T)	*Bacillus aryabhattai* strain B8W22	NR_115953	100
11022(K)	*Bacillus toyonensis* strain BCT-7112/*B. thuringiensis* strain IAM 12077	NR_121761.1 NR_043403.1	99.93
10222(K)	*Brevibacillus formosus* strain DSM 9885	NR_040979.1	99.57
29720(N)	*Psychrobacillus psychrodurans* strain 68E3	NR_025409	99.93
31320(K)	*Bacillus frigoritolerans* comb. nov. strain DSM 8801	NR_115064	100
31120(K)	*Bacillus frigoritolerans* comb. nov. strain DSM 8801	NR_117474.1	99.93
30020(N)	*Bacillus frigoritolerans* comb. nov. strain DSM 8801	NR_115064	100
30420(T)	*Bacillus mobilis* strain MCCC 1A05942	NR_157731	100
9922(K)	*Bacillus subtilis* strain DSM 10	NR_027552.1	100
31520(K)	*Bacillus subtilis* strain DSM 10	NR_027552	100
31420(K)	*Bacillus pumilus* strain ATCC 7061	NR_043242.1	100
30620(T)	*Bacillus pumilus* strain ATCC 7061	NR_043242	99.86
29820(N)	*Bacillus pumilus* strain NBRC 12092	NR_112637	100
30820(T)	*Bacillus pumilus* strain NBRC 15720 = DSM 1321	NR_112726	99.79
30320(T)	*Bacillus pumilus* strain NBRC 12092	NR_112637	100
30520(T)	*Bacillus pumilus* strain NBRC 12092	NR_112637	100
10322(K)	*Brevibacillus laterosporus* strain DSM 25	NR_112212.1	99.78
31220(K)	*Bacillus thuringiensis* strain IAM 12077	NR_043403	100
8722(K)	*Janthinobacterium lividum* strain DSM 1522	NR_026365.1	99.93
30120(N)	*Arthrobacter bambusae* strain THG-GM18	NR_133968	99.69

## Data Availability

The metabarcoding read data reported in this study were deposited in GenBank under the study accession No. PRJNA940644. The final dataset data of 16S rRNA metabarcoding are presented in the Appendix A. Other raw data from this study will be provided by the authors upon request, without restrictions.

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
