# Peer review of "Links between Soil Bacteriobiomes and Fungistasis toward Fungi Infecting the Colorado Potato Beetle"

_microorganisms, 2023, doi:10.3390/microorganisms11040943_

Round 1

Reviewer 1 Report

The manuscript “Links between soil bacteriobiomes and fungistasis toward  fungi infecting the Colorado potato beetle" reports This paper provided critical evidence of the relationship between soil, fungi, and insects, as well as how this interaction affects CPB mortality. Interestingly, the author has used several techniques to support this study, which deserves publication. However, some issues need to be explained or revised before this manuscript can be accepted.

Comments and Suggestions for Authors

The abstract is poorly written; we suggest rewriting the abstract to include a proper introduction, materials and methods, results, and proper conclusions drawn by this study. The grammar is very poor in the abstract section.

Line 16-18 write full name of Beauveria and Metarhizium

Line 22 write “in an assay”. There are many types of assays used in experiments. Mention the name of assay.

Line 23 use beetle instead of “beetles” 

Line 28-28 “16S rDNA; metabarcoding; insect pathogen; ascomycetes; Leptinotarsa  decemlineata; agricultural practice” all of these words are not present in the abstract Consider the rewriting of keywords. Also, Leptinotarsa decemlineata is not in italic font.

The Colorado Potato Beetle is mentioned in the abstract, but the scientific name is not written in the abstract. Consider it.

Introduction.

Iine 49-51. What is the meaning of “(e.g., [7]; [8])”. Consider rewriting with appropriate manner.

The font size is inconsistent throughout the manuscript. E.g line 42, line 51-52 …………..

Line 58 “(reviewed by [11]; [12]).” Consider rewriting this typo in the whole manuscript.

What is the role of “mycoses” in insects?

The in text citation style is accurate and inconsistence.

Objectives of the study are not clear.

Materials and Methods

What does author want to say by “Finishing feeding larvae of the CPB”? Please elaborate

Line 135-137. In the fields, there are lot of spray and other pesticide effects are present for long time. How it is possible that the larvae collected were from the fields that are free from the application of biological insecticides? Does after collection the larvae of CPB was reared for some generation before experiments?

What are the authentic source of collection of the strains used in the experiments?

Results:

Figure 1 from the graph A and B, not clear that which graph represents the soils of kitchen potato gardens and conventional potato field? Specify.

Figure 2 has also the same problem as the figure 1

Why this study only focus the CPB larvae of finishing feeding stage?

Discussion

The start of the discussion is rather poor. However, the author has effectively described and compared the results with the other studies.

Line 525-527. Is there is any reference for your logic. If yes, please mention.

The author used finishing feeding stage larvae in this study. Possibly this is the larva that was going into the soil for pupation. If yes, then there is an effect of metamorphosis (pupation) on the mortality of insects. How did the author get the appropriate results of mortality while keeping in mind the effect of metamorphosis (pupation) on mortality in insects?

The conclusions drawn by this study are very clear and excellent.

Author Response

Comment. The manuscript “Links between soil bacteriobiomes and fungistasis toward  fungi infecting the Colorado potato beetle" reports This paper provided critical evidence of the relationship between soil, fungi, and insects, as well as how this interaction affects CPB mortality. Interestingly, the author has used several techniques to support this study, which deserves publication. However, some issues need to be explained or revised before this manuscript can be accepted.

Response. Thank you for work with our paper and valuable comments!

Comment. The abstract is poorly written; we suggest rewriting the abstract to include a proper introduction, materials and methods, results, and proper conclusions drawn by this study. The grammar is very poor in the abstract section.

Response. Abstract rewritten. Now it is looks as: Entomopathogenic fungi can be inhibited by different soil microorganisms, but the effect of a soil microbiota on fungal growth, survival, and infectivity toward insects are insufficiently understood. We investigated the level of fungistasis toward Metarhizium robertsii and Beauveria bassiana in soils of potato conventional fields and potato kitchen gardens. Agar diffusion methods, 16S rDNA metabarcoding, bacterial DNA quantification, and assays of Leptinotarsa decemlineata survival in soils inoculated with fungal conidia were used. Soils of kitchen gardens showed stronger fungistasis toward M. robertsii and B. bassiana, and at the same time highest density of the fungi compared to soils of conventional fields. The fungistasis level depended on the quantity of bacterial DNA and relative abundance of Bacillus, Streptomyces, and some Proteobacteria, whose abundance levels were the highest in kitchen garden soils. Cultivable isolates of bacilli exhibited antagonism to both fungi in vitro. Assays involving inoculation of nonsterile soils with B. bassiana conidia showed trends toward elevated mortality of L. decemlineata in highly fungistatic soils compared to low-fungistasis ones. Introduction of antagonistic bacilli into sterile soil did not significantly change infectivity of B. bassiana toward the insect. The results support the idea that entomopathogenic fungi can infect insects within a hypogean habitat despite high abundance and diversity of soil antagonistic bacteria.

Comment. Line 16-18 write full name of Beauveria and Metarhizium

Response. We did not molecular identification of Beauveria and Metarhizium in assay of CFU count. Each genera includes several cryptic species in temperate zone of Palearctic (DOI: 10.3852/10-302, DOl: 10.3852/07-202). More widespread are B. bassiana and B. pseudobassiana, M. robertsii and M. brunneum (doi:10.1016/B978-0-12-821237-0.00003-2). We are sorry, we can not list all species  because this will overload and complicate the text.

Comment. Line 22 write “in an assay”. There are many types of assays used in experiments. Mention the name of assay.

Response. Corrected. Please, see respose to comment 2. 

Comment. Line 23 use beetle instead of “beetles”  

Response. Only Latin name is in abstract now

Comment. Line 28-28 “16S rDNA; metabarcoding; insect pathogen; ascomycetes; Leptinotarsa  decemlineata; agricultural practice” all of these words are not present in the abstract Consider the rewriting of keywords. Also, Leptinotarsa decemlineata is not in italic font.

Response. Corrected: soil microbiota; fungistasis; entomopathogenic fungus; Beauveria; Metarhizium; Leptinotarsa decemlineata; potato field; agricultural practice

Comment. The Colorado Potato Beetle is mentioned in the abstract, but the scientific name is not written in the abstract. Consider it.

Response. Only Latin name is in abstract and only English name in the title now

Comment. Introduction. Iine 49-51. What is the meaning of “(e.g., [7]; [8])”. Consider rewriting with appropriate manner.

Response. Corrected throughout the text

Comment. The font size is inconsistent throughout the manuscript. E.g line 42, line 51-52

Response. Corrected.

Comment. Line 58 “(reviewed by [11]; [12]).” Consider rewriting this typo in the whole manuscript.

Response. Corrected throughout the text

Comment. What is the role of “mycoses” in insects?

Response. Information added: These fungi can cause different levels of mortality in insect populations and are used as biocontrol agents against crop pests [7, 8].

Comment. The in text citation style is accurate and inconsistence.

Response. Corrected though entire text

Comment. Objectives of the study are not clear.

Response. Text on objectives rewritten: The objective of the study was to discover possible relations between the level of fungistasis in soils of potato fields, the structure of the bacterial microbiota, and the development of fungal infections in the CPB during metamorphosis, in particular, 1) to examine the level of fungistasis and the CFU count of Beauveria and Metarhizium fungi in soils of potato fields subjected to different agricultural practices; 2) to analyze  bacterial communities of the soils by metabarcoding sequencing of the 16S rRNA gene; 3) to isolate cultivable bacteria and assess their antagonistic activity against M. robertsii and B. bassiana; and 4) to evaluate the effect of soils with different fungistasis levels on the development of fungal infections in the CPB during metamorphosis as well as the impact of the introduction of antagonistic bacteria into the soil on the mortality of the CPB from mycoses.

Comment. Materials and Methods What does author want to say by “Finishing feeding larvae of the CPB”? Please elaborate

Response. Information added: Finishing feeding larvae of the CPB L. decemlineata (6–7 days postmolt in IV instar) served as test insects.

Comment. Line 135-137. In the fields, there are lot of spray and other pesticide effects are present for long time. How it is possible that the larvae collected were from the fields that are free from the application of biological insecticides? Does after collection the larvae of CPB was reared for some generation before experiments?

Response. Of course, we cannot exclude the complete absence of chemical insecticides. However, as our practice shows, a whole laboratory generation leads to some weakening of insects, in particular, a lower weight of larvae, compared with those collected in the fields. In all our experiments with insects, we had appropriate controls in which no death with insecticidal symptomatic was observed, but only death with symptoms of mycosis. We have added this information to the results section 3.5, 3.6. Please see the revised manuscript.

Comment. What are the authentic source of collection of the strains used in the experiments?

Response. Information added in subsection 2.2.: P-72 was isolated from the CPB in Latvia in 1972; Sar-31 was isolated from Calliptamus italicus in Karasuk district (Novosibirsk Oblast, Western Siberia) in 2001.

Comment. Results: Figure 1 from the graph A and B, not clear that which graph represents the soils of kitchen potato gardens and conventional potato field? Specify.

 Response. Specified in figure legend as required.

Comment. Figure 2 has also the same problem as the figure 1

Response. Specified in figure legend as required

Comment. Why this study only focus the CPB larvae of finishing feeding stage?

Response. Finishing feeding larvae, and then pupae and young adults are contacted with soil and infected by fungi.  In experiments, we incubated CPB in soils during whole period of metamorphosis. Explanations added in Introduction (last paragraph), M&M (2.9, 2.10) and discussion sections. Please, see revised version of MS.    

Comment. Discussion The start of the discussion is rather poor. However, the author has effectively described and compared the results with the other studies.

Response. First paragraph of Discussion modified: Here, using potato field soils, we show relations between the level of fungistasis toward entomopathogenic fungi, the bacterial communities, and infectivity of B. bassiana toward the CPB during metamorphosis. Antagonistic effects on entomopathogenic fungi were more pronounced in kitchen garden soils (Karasuk) than in soils of conventional fields (Novosibirsk), in agreement with the high quantity of total bacterial DNA and high relative abundance of Firmicutes (Bacilli), Proteobacteria, and certain taxa of Actinobacteria (Streptomyces) in kitchen garden soils. Nevertheless, soils with higher fungistasis contained higher CFU counts of Beauveria and Metarhizium. Moreover, we registered trends of elevated infectivity of B. bassiana toward CPB in highly fungistatic soils compared to low-fungistasis ones.

 Comment. Line 525-527. Is there is any reference for your logic. If yes, please mention.

 Response. Please, see explanations of these assumptions in following sentences with references 60 and 61.

Comment. The author used finishing feeding stage larvae in this study. Possibly this is the larva that was going into the soil for pupation. If yes, then there is an effect of metamorphosis (pupation) on the mortality of insects. How did the author get the appropriate results of mortality while keeping in mind the effect of metamorphosis (pupation) on mortality in insects?

Response. All insects in the experiments passed process of metamorphosis. At the same time, death was not observed in the corresponding controls. Explanations added in introduction section (last paragraph) methods (2.9, 2.10) and results (3.5., 3.6). Please, see revised manuscript

Comment. The conclusions drawn by this study are very clear and excellent.

Response. Thank you for constructive comments!

Reviewer 2 Report

The paper of Chertkova et al entitled “Links between soil bacteriobiomes and fungistasis toward fungi infecting the Colorado potato beetle” represents a quite nice contribution to the understanding of the complex interaction between soil bacteria and entomopathogenic fungi that can infect Colorado potato beetle. Overall, the experiments were conducted in a satisfactory manner and the paper is well written.

It is quite important that more evidence is provided regarding the fact that fungistasis is not necessary associated with reduced ability of a pathogen to infect a host.

In several instances in the formatting of the text is not OK. For example, in many places some words appear written with larger fonts. For example, check lines 51, 91, 221, 237, 240, 247, 311, 337, 442, 539, 514 and 548. Maybe I missed a few.

Species names should be italicized in lines 255, 260 and 262.

Line 182. It is quite unusual to start a paragraph with “This was done”; please revise this sentence.

In lines 157 and 300, 16 S rDNA got transformed in 2.5.16 and 3.2.16, respectively,

In Lines 220, 221 and 237 the order in which information regarding the manufacturer should be revised; it should be city, state, and country.

Author Response

Comment. The paper of Chertkova et al entitled “Links between soil bacteriobiomes and fungistasis toward fungi infecting the Colorado potato beetle” represents a quite nice contribution to the understanding of the complex interaction between soil bacteria and entomopathogenic fungi that can infect Colorado potato beetle. Overall, the experiments were conducted in a satisfactory manner and the paper is well written. It is quite important that more evidence is provided regarding the fact that fungistasis is not necessary associated with reduced ability of a pathogen to infect a host.

Response. Thank you for work with paper and valuable comments!

Comment. In several instances in the formatting of the text is not OK. For example, in many places some words appear written with larger fonts. For example, check lines 51, 91, 221, 237, 240, 247, 311, 337, 442, 539, 514 and 548. Maybe I missed a few.

Response. Corrected.

Comment. Species names should be italicized in lines 255, 260 and 262.

Response. Corrected.

Comment. Line 182. It is quite unusual to start a paragraph with “This was done”; please revise this sentence.

Response. Corrected: This quantification was performed by real-time PCR on a CFX96 Touch real-time PCR detection system (Bio-Rad, Hercules, CA, USA).

Comment. In lines 157 and 300, 16 S rDNA got transformed in 2.5.16 and 3.2.16, respectively,

Response. Corrected.

Comment. In Lines 220, 221 and 237 the order in which information regarding the manufacturer should be revised; it should be city, state, and country.

Response. Corrected.